# Entangled Motifs in Membrane Protein Structures

**DOI:** 10.3390/ijms24119193

**Published:** 2023-05-24

**Authors:** Leonardo Salicari, Antonio Trovato

**Affiliations:** 1Department of Physics and Astronomy ‘Galileo Galilei’, University of Padova, Via Marzolo 8, 35031 Padova, PD, Italy; 2National Institute of Nuclear Physics (INFN), Padova Section, Via Marzolo 8, 35131 Padova, PD, Italy

**Keywords:** membrane proteins, entanglement, chirality, co-translational folding

## Abstract

Entangled motifs are found in one-third of protein domain structures, a reference set that contains mostly globular proteins. Their properties suggest a connection with co-translational folding. Here, we wish to investigate the presence and properties of entangled motifs in membrane protein structures. From existing databases, we build a non-redundant data set of membrane protein domains, annotated with the monotopic/transmembrane and peripheral/integral labels. We evaluate the presence of entangled motifs using the Gaussian entanglement indicator. We find that entangled motifs appear in one-fifth of transmembrane and one-fourth of monotopic proteins. Surprisingly, the main features of the distribution of the values of the entanglement indicator are similar to the reference case of general proteins. The distribution is conserved across different organisms. Differences with respect to the reference set emerge when considering the chirality of entangled motifs. Although the same chirality bias is found for single-winding motifs in both membrane and reference proteins, the bias is reversed, strikingly, for double-winding motifs only in the reference set. We speculate that these observations can be rationalized in terms of the constraints exerted on the nascent chain by the co-translational bio-genesis machinery, which is different for membrane and globular proteins.

## 1. Introduction

Membrane proteins are present in and interact with the plasma, nuclear, and organelle membranes of all living organisms. Their structural and functional properties affect the structure and composition of the host membranes and how they react to changes in the surrounding environment. Membrane proteins are relatively abundant (25–30% of all proteins) and play an essential role in a wide range of cellular processes, from active transport to signal transduction, to name but a few [1,2]. Despite this, considerably less information is available about the structures and functions of membrane proteins with respect to other protein classes. Obtaining high-resolution structures for membrane proteins was extremely challenging until the last decade [3]. Technological advances [4], however, have led to a surge in the determination of membrane protein structures with atomic resolution in recent years. More than 1500 of them are now available [5]; this number is much lower than the almost 200,000 protein structures deposited in the Protein Data Bank [6] by the end of 2022, yet is large enough to allow systematic surveys of general structural properties of membrane proteins. In particular, in this contribution, we wish to conduct a survey of the presence of entangled motifs in membrane protein structures. In our study, we will consider and compare the properties of both monotopic and transmembrane proteins. In this way, we can assess whether the need to connect both sides of the membrane affects the presence and properties of entangled motifs.

A large body of research showed that the kinetic and thermodynamic features of the folding process of globular proteins are encoded in simple topological descriptors of their native state structures, even within a coarse-grained description [7,8,9], such as, for example, the properties of the network of residues in contact with each other [10,11,12]. Structure-based energy functions, whose global minimum is attained for the native structure, thus became popular and successful tools for predicting folding mechanisms [13,14]. Structure-based models are widely used to investigate the folding process of membrane proteins as well [15,16,17].

The organization of the network of native contacts does not determine, however, the topology of the protein backbone chain as a continuous curve in three-dimensional space. For example, the same set of native contacts could in principle be compatible with either a knotted or an unknotted topology. The discovery of knots in a few proteins [18] was a surprise because they seem to pose a big challenge to the folding process. Their presence was suggested to be related to some biological function or stability requirement [19,20], and the mechanisms which thread the protein backbone to form knots are being carefully investigated [21,22,23,24,25]. Notably, transmembrane proteins with a knotted topology were also detected [25] and the evolutionary origin of a slipknotted motif in a monovalent cation–proton antiporter was discussed [26].

Recently, it was shown that motifs other than knots may lead to protein structures with non-trivial spatial arrangements. These include knotoids [27], slipknots [28], lassos [29,30], pokes [31] and non-covalent lassos detected by Gaussian entanglement [32,33,34,35]. The latter is a generalization of the mathematical concept of linking number [36] to discretized and possibly open curves, quantified by means of suitably modified Gauss integrals [32,37,38,39]. A crucial question is whether and how these topologically entangled motifs affect the protein energy landscape and the folding process [40,41]. For example, the Gaussian entanglement was found to be significantly correlated with the “in vitro” folding rate [33] for globular proteins, so that a higher degree of entanglement slows down the folding process.

Entangled loops are looped lasso-like segments of a protein chain that display large Gaussian entanglement when threaded by another segment, possibly not looped, of the same protein chain. Recently, it was discovered that they are present in roughly one-third of known single-domain proteins [35], much more than knots [19]. Moreover, the amino acids at the ends of a loop interact with each other with significantly weaker energy, on average, if the loop is entangled [35]; this is possibly the consequence of an evolutionary pressure to avoid kinetic traps [42]. Entangled loops were also found to be distributed asymmetrically with respect to the chain portion they are entangled with, the thread. The latter is located more frequently on the N-terminal side of the loop [35]. In the context of co-translational folding [43], this implies that entangled loops are synthesized at the ribosome and hence folded, on average, later than the thread. A significant preference for N-thread-entangled motifs with positive chirality was also detected [35]—an observation that has yet to be explained.

Very recently, a striking connection was established between the presence of misfolded protein sub-populations during and just after protein translation and their entanglement properties. For example, an abundance of entangled motifs characterizes a subset of proteins prone to misfolding and aggregation under heat stress when newly synthesized but not once matured [44]. This may suggest that these proteins rely more on the protein homeostasis machinery to reach their native states. Along the same lines, using coarse-grained structure-based models of protein translation, it was predicted that one-third of proteins can misfold into soluble less-functional states, that bypass the protein homeostasis network, avoiding aggregation and rapid degradation [45]. Such misfolded species were characterized as long-lived kinetic traps, native-like in several respects, with misfolding due to a change in the entanglement properties of the native structure. Moreover, the shift in competition between differently entangled misfolded subpopulations, along cotranslational folding pathways, was shown to determine the functional impact of synonymous mutations [46].

In this contribution, we use the Gaussian entanglement technique to detect and characterize entangled loops in the currently available data sets of membrane protein structures. Naively, we could expect to find much less entangled motifs in membrane proteins with respect to globular proteins; the task of folding correctly large multipass membrane proteins with several transmembrane segments is already difficult enough without the additional requirement of forming the correct entangled motifs. Surprisingly, although reduced with respect to the 32% observed for general proteins, we find a significant fraction of entangled motifs in transmembrane proteins as well: 20% of transmembrane protein domains (as identified by CATH [47]) host at least one entangled loop, whereas monotopic proteins behave more similarity to general proteins. Several cases of highly entangled membrane proteins, with multiple windings of the thread around the entangled loops and/or with coexisting entangled motifs with opposite chiralities are identified and discussed. Moreover, the preference for N-thread entangled motifs with positive chiralities is observed for transmembrane proteins in a similar way to general proteins. On the other hand, the chirality preference for multiple wound-entangled motifs is found to be different for transmembrane proteins. We speculate that this may be rationalized in the context of the different cotranslational machineries involved in the bio-genesis of membrane or globular proteins. Overall, our results imply that entangled motifs are commonly found in membrane proteins, raising the question of their possible biological significance. Finally, the possible relevance at a cellular level of membrane protein mis-entangled sub-populations that are otherwise native-like and therefore bypass the protein homeostasis network, is also called into question.

## 2. Results

### 2.1. Pipeline for Computational Analysis

We compiled a data set of experimentally resolved protein chain structures that have been annotated to interact with one of the membranes present in the cell (see Section 4.1 for details). In summary, we collected information from MemProtMD [48], mpstruc [5], OPM [49,50] and PDBTM [51,52], using the RCSB Protein DataBank [6,53] as an aggregator. Chains are classified as transmembrane or monotopic and, in parallel, as peripheral or intrinsic. In practice, the two ways of classifying membrane proteins almost always coincide: almost all monotopic ones are peripheral and all intrinsic are transmembrane (see Table 1). At this stage, we consider different chains from the same complex as separate entries in the database. Note that the entangled motif here could, in principle, intermingle two different chains in the same complex, as observed in the case of domain-swapped dimers [32]. However, such an analysis would be hardly feasible due to the need to check the possible entanglement for all choices of the two portions from different chains (see Section 4.2).

In order to compare the results of our survey of entangled motifs with a previous analysis [35], and to save computational time in the present one, we split each chain in our data set into different domains, using the CATH database [47]. Entanglement is not expected to occur, however, between chain portions in different domains. To reduce redundancy in the data set, a 35% homology filter is applied. Note that CATH single domains are typically larger than the transmembrane domains used to refer to in the context of membrane proteins. At this stage, we are considering different domains from the same chain as separate entries.

Since backbone continuity of the chain portions is essential to assess their mutual entanglement, domains composed of multiple fragments, namely composed of multiple non-consecutive chain portions, and with missing regions longer than 10 amino acids, are also excluded. Shorter missing regions are reconstructed with the  MODELLER software [54,55]. After discarding error-prone PDB files, the final dataset contains 1378 domains. Where, 871 are classified as transmembrane, 494 are monotopic, 871 are intrinsic and 490 are peripheral. Note that all such labels are originally attached to the whole chain, so that different domains in our data set, originating from the same chain, will all carry the same label. Appendix A summarizes the overall pipeline.

### 2.2. Membrane Proteins Contain a Significant Fraction of Entangled Motifs

The entanglement properties of the data set of membrane protein domains built in this work (see Section 2.1) will generally be benchmarked against a reference set of 16,709 protein domain structures that were selected in [35] (see Section 4.1 for details).

The following procedure is based on the evaluation of different entanglement indicators. In a given structure, for each close contact between two different residues (see Section 4.1 for a detailed definition of close contacts), we consider the chain portion looping between the two residues (the “loop”) and evaluate its entanglement with all other chain portions (the “threads”, in general not looped) non overlapping with it. The entanglement of a given loop–thread pair is quantified by the $G^{\prime }$ indicator (see Section 4.2 for more details).

$G^{\prime }$ is an integer number for two closed curves, being zero for non-concatenated curves. Its generalization to open curves yields non-integer numbers, to detect entangled loop–thread motifs when $\left|G^{\prime }\right|\geq 1$ [35]. The sign of $G^{\prime }$ detects the chirality of the mutual winding of the two curves if a direction is defined along them (as is the case for protein chains from the N- to the C-terminus). The $\left|G^{\prime }\right|=1$ threshold for entanglement detection is a strict one. More recently, a less strict threshold $\left|G^{\prime }\right|=0.5$ was used in other contexts, to estimate the difference in entanglement with respect to the native structure [40,45,46], or to build a reaction coordinate to describe the folding process [41]. For each domain structure, we eventually select the loop–thread pair with the maximum value of $\left|G^{\prime }\right|$ and use the corresponding value, $G_{\max}^{\prime }$, as an indicator of the overall entanglement of the domain.

The distribution of the $G_{\max}^{\prime }$ values for all 1378 membrane protein domains is shown in Figure 1a, together with the same distribution obtained for the reference set of 16,709 protein domain structures. The main features of the entanglement indicator distributions are common to both sets: an overall symmetry for chirality change is roughly present; the peaks/shoulders at $\left|G_{\max}^{\prime }\right|≃0.2÷0.3$ correspond to non-entangled structures; the peaks at $\left|G_{\max}^{\prime }\right|≃1$ signal the significant presence of entangled structures; the distribution tails for $\left|G_{\max}^{\prime }\right|≳1.5$ are appreciably populated; “extreme” values of $\left|G_{\max}^{\prime }\right|≃2$, which detect deeply entangled structures where the loop and the thread wind around each other twice, are observed. The same general trend is common to the subsets of membrane protein domains corresponding to the most represented organisms (see Figure 1b) if one allows for fluctuations due to the smaller statistics available for each subset. We observe that the values of $\left|G_{\max}^{\prime }\right|≃0$ are depleted in general for non-entangled structures because the indicator is maximized over all loop–thread pairs in a given structure.

However, a couple of features differentiate the entanglement properties of membrane protein domains with respect to the reference set. For positive chirality, the peak at $G_{\max}^{\prime }≃0.2÷0.3$ for non-entangled structures is enhanced in membrane proteins, at the expense of the peak at $G_{\max}^{\prime }≃+1$ (see Figure 1a). Furthermore, for deeply entangled structures with $\left|G_{\max}^{\prime }\right|≳1.5$, a bias favoring negative over positive chirality is present only in the reference set.

A better insight can be gained by looking at the distribution of the overall entanglement indicator separately for monotopic and transmembrane protein domains (see Figure 2a). The enrichment in non-entangled structures with positive chirality ($G_{\max}^{\prime }≃0.2÷0.3$) is due to transmembrane proteins and can be traced to the smallest domains that host only one transmembrane helix. These single-pass transmembrane proteins are most naturally not entangled. Furthermore, the distribution for monotopic proteins displays two features that make it more similar to the reference case (see Figure 1a) than to the transmembrane case; namely, the relative height of the peaks for non-entangled ($G_{\max}^{\prime }≃0.2÷0.3)$ and entangled ($G_{\max}^{\prime }≃+1$) structures with positive chirality and the populated tail for deeply entangled structures with negative chirality ($G_{\max}^{\prime }<-1.5$).

The entanglement properties of monotopic domains are more similar to the reference set than those of transmembrane ones. This is also confirmed by looking at the entanglement survival function in Figure 2b, where we plot how the fraction of structures in the data set with values of the entanglement indicator larger than $\left|G_{\max}^{\prime }\right|$ decreases as the latter is increased. The transmembrane proteins are, in general, less entangled than monotopic ones, which in turn are less entangled with respect to the reference set. The more a protein interacts with the membrane, the more constraints are imposed on the presence of entangled motifs, as could have been expected a priori. On the other hand, the fraction of “survived” structures at $\left|G_{\max}^{\prime }\right|=1$ is still significant, being 0.20 for transmembrane proteins and 0.27 for monotopic ones, against 0.32 of the reference set. One-fifth of the transmembrane protein domains in our data set host an entangled motif.

From a polymer physics perspective, the presence of entangled motifs is generally expected to increase with the length of the chain [22]. The distributions of domain lengths in the different data sets that we compare are shown in Figure 2c. An enrichment in small size domains is visible for transmembrane proteins, due to the presence of the single-pass domains discussed above. However, the three distributions are overall similar to each other. This ensures that the above conclusions on the comparison of the entanglement properties of the three data sets are not biased by different domain–length distributions.

### 2.3. A Specific Chirality bias for Double Winding Entangled Motifs Characterizes Membrane Proteins

We now analyze entanglement properties focusing on the difference between N-threads (loop-thread pairs in which the thread is on the N-terminal side with respect to the loop) and C-threads (loop-thread pairs in which the thread is on the C-terminal side with respect to the loop). A small but statistically significant preference for entangled N-threads was previously uncovered in the reference set of protein domain structures, originating from entangled N-threads with positive chirality and $G^{\prime }≃+1$ [35]. This result connects the presence of entangled motifs in protein structures with the properties of co-translational folding.

Here, we associate two separate indicators, $G_{N}^{\prime }$ and $G_{C}^{\prime }$, to each structure in the data sets. $G_{N}^{\prime }$ is the $G^{\prime }$ value of the N-thread with the maximum $\left|G^{\prime }\right|$, whereas $G_{C}^{\prime }$ is the $G^{\prime }$ value of the C-thread with the maximum $\left|G^{\prime }\right|$. The overall entanglement indicator $G_{\max}^{\prime }$ investigated in Section 2.2 obeys $\left|G_{\max}^{\prime }\right|=\max\left\{\left|G_{N}^{\prime }\right|,\left|G_{C}^{\prime }\right|\right\}$. The overall entanglement will be also referred to as an N-thread (C-thread) if $\left|G_{N}^{\prime }\right|>\left|G_{C}^{\prime }\right|$ ($\left|G_{C}^{\prime }\right|>\left|G_{N}^{\prime }\right|$), reflecting the nature of the most entangled loop–thread pair in the structure.

In Figure 3, we show the scatter plot of the $G_{N}^{\prime }$ and $G_{C}^{\prime }$ values for all membrane protein domains in our data set. Data points cluster around the diagonal ($G_{C}^{\prime }=G_{N}^{\prime }$) and the anti-diagonal ($G_{C}^{\prime }=-G_{N}^{\prime }$) lines in the ($G_{N}^{\prime }$,$G_{C}^{\prime }$) plane. In a given structure, the most entangled motifs of both kinds (N-thread or C-thread) are likely to be entangled to similar extents. Points around the diagonal refer to structures for which the most entangled N-thread and the most entangled C-thread are characterized by the same chirality. Points around the anti-diagonal refer to structures for which the most entangled N-thread and the most entangled C-thread are characterized by opposite chiralities. The latter case occurs less frequently. In summary, different entangled motifs may coexist in membrane proteins, even with opposite chiralities.

To investigate in more detail, the relationship between the kind and chirality of entangled motifs, we partition the “survived” structures at a given value of $\left|G_{\max}^{\prime }\right|$ (see Figure 2b) into four different groups to take into account all combinations of chirality and kinds of the most entangled motif. In these groups, we collect only structures from the diagonal quadrants in Figure 3, whose $G_{N}^{\prime }$ and $G_{C}^{\prime }$ values share the same sign. Corresponding groups could be defined for structures in the anti-diagonal quadrants, with opposite signs for $G_{N}^{\prime }$ and $G_{C}^{\prime }$, but their low statistics would not allow significant conclusions. On the other hand, grouping together structures from diagonal and anti-diagonal quadrants, with different topological arrangements of entangled motifs, would not be appropriate.

In Figure 4, we show how the fraction of “survived” structures in the four groups (“grouping fractions”) changes with increasing entanglement level $\left|G_{\max}^{\prime }\right|$ for the reference, monotopic and transmembrane domain data sets. As the number of “survived” structures decreases with $\left|G_{\max}^{\prime }\right|$, the statistical error increases and becomes rapidly huge, especially for the membrane protein data sets. Despite this, some robust trends emerge, which are consistent with the observations already made in Section 2.2 and with previous work [35].

The four groups are not populated uniformly at low $\left|G_{\max}^{\prime }\right|$ for non-entangled structures. The hierarchy of the four groups is only roughly preserved in the different data sets. The local geometrical properties of secondary structure elements and, therefore, the secondary structure content of the different data sets, are possibly a relevant factor in this respect. More importantly, as $\left|G_{\max}^{\prime }\right|$ approaches the entanglement threshold $\left|G_{\max}^{\prime }\right|≃1$, the fraction of N-threads with positive chiralities (with $G_{N}^{\prime }>G_{C}^{\prime }>0$) is consistently enriched in all data sets. N-threads that wind once with positive chirality around the loop they entangle with ($1≲G_{\max}^{\prime }≲1.5$) are enriched for both membrane protein data sets in a similar way to the reference set. Note that a similar result for the reference set had already been obtained by analyzing the set of all entangled loops [35], where many of those were included for a given structure. Here, instead, we collect only one motif per structure.

Strikingly, the behavior of the grouping fractions discriminate between the reference and the transmembrane sets when $\left|G_{\max}^{\prime }\right|≳1.5$, that is, for entangled motifs in which the thread winds twice around the loop. In the reference set, there is a sudden change in the favored chirality; double winding motifs are clearly enriched for negative chiralities (both N-threads and C-thread), whereas the fraction of N-threads with positive chirality drops down. This change in the favored chirality is not seen for transmembrane proteins and is seen to some extent (but possibly only because of small statistics) for monotopic proteins. The effect of the presence of the membrane is visible only for double winding motifs, and to a much greater extent for transmembrane protein domains than for monotopic ones. Note that the enrichment in double-winding motifs with negative chiralities for the reference set is consistent with what was already observed about the negative $G_{\max}^{\prime }<0$ tails of the distributions in Figure 1a. On the other hand, the ranking of the data sets according to the strength of the “membrane effect” is consistent with what was observed in Figure 2.

### 2.4. Membrane Proteins Can Self-Entangle in a Variety of Ways

We now show and discuss briefly a couple of specific examples of entangled membrane proteins identified in this work. They are all taken from the transmembrane protein data set.

In Figure 5a, we show the CATH domain 4P02A02. This is an example of a positive chirality N-thread single winding with $G_{N}^{\prime }=1.10>G_{C}^{\prime }=0.79$. A temptative planar structure projection that helps visualize the winding of the thread around the loop is provided showing the approximate position of secondary structure elements. 4P02 is a bacterial cellulose synthase (Bcs) complex [56], annotated as a membrane protein by MemProtMD and PDBTM, and as a transmembrane one by OPM. It consists of two chains associated with the inner membrane, the cellulose synthase catalytic subunit (BcsA), which contains most of the transmembrane segments, and the cyclic di-GMP-binding protein (BcsB), which has one C-terminal transmembrane segment across the inner membrane and is anchored to the periplasmic membrane at its N-terminal. CATH lists two different cytosolic domains (4P02A01 and 4P02A02) for BcsA. The domain 4P02A02, found to be entangled in our analysis, contains 121 residues and the entangled motif contains both α and β secondary structure elements. This example illustrates pointedly the caution with which any large-scale analysis of membrane proteins should be taken; 4P02A02 is labeled as a transmembrane domain in our data set because the protein chain that contains the domain is indeed transmembrane. However, 4P02A02 is a cytosolic domain and the hosted entangled motif is not involved in any direct interaction with the membrane in the native PDB structure. At the same time, the folding process involves the whole chain and therefore the insertion of all transmembrane segments, so that the folding of the cytosolic domain may still be influenced by the presence of the membrane.

In Figure 5b, we show the CATH domain 3GIAA00. This is an “extreme” example of a positive chirality C-thread double winding with $G_{N}^{\prime }=2.32<G_{C}^{\prime }=2.39$. A temptative planar structure projection that helps to visualize the double winding of the thread around the loop is provided, showing the approximate position of the secondary structure elements. 3GIA is a single chain APC transporter [57], annotated as a membrane protein by MemProtMD and PDBTM, and as a transmembrane one by OPM and mpstruc. It is basically a unique CATH domain (483 residues) consisting of 12 transmembrane helical segments, with both the N-terminal and C-terminal on the cytosolic side. In this example, the entangled motif is fully transmembrane and involves only an α-helical secondary structure.

As a last example, in Figure 6, we show the CATH domain 4MT4A00. This is an example of the coexistence of single winding entangled motifs with opposite chiralities, with a positive chirality N-thread ($G_{N}^{\prime }=1.20$) and a negative chirality C-thread ($G_{C}^{\prime }=-1.19$). The cartoon structure and the temptative planar projection that helps visualize the winding of the thread around the loop are shown separately for each entangled motif. The threaded loops are different in the two motifs, yet partially overlap. 4MT4 is a CmeC outer membrane channel [58] from *Campylobacter jejuni*, annotated as a membrane protein by MemProtMD and PDBTM, and as a transmembrane one by OPM and mpstruc. It is a homo-trimer in which each chain corresponds to a unique CATH domain (472 residues) and contributes four β-strands and six α-helices to form the transmembrane β-barrel domain and the periplasmic α-helical domain. Moreover, an equatorial domain, made up of four short α-helices from each chain, surrounds the middle section of the periplasmic α-helical domain (in this description domains refer to the trimer complex). In this example, both entangled motifs span a large portion of the protein chain, including both the transmembrane and the periplasmic domains. In the N-thread motif, the entangled loop is closed by a non-covalent contact (yellow in Figure 6) within the periplasmic domain, whereas, in the C-thread motif, the entangled loop is closed by a non-covalent contact in the β-barrel domain.

## 3. Discussion

Entangled loops, that is, chain portions closing like a lasso around a second threading segment by means of a non-covalent contact, were found to appear in one-third of protein domains [35]. Threads were found significantly more often at the N-terminal side of entangled loops, pointing to a connection between the formation of entangled motifs and co-translational folding [35]. In a computational study, we recently showed how the presence of entangled motifs in the native structure might deeply affect the folding mechanisms; a long-lived kinetic trap is formed when the thread is not folded properly prior to loop closure, even in the case of a small globular protein [41].

In this contribution, we surveyed known membrane protein structures to investigate the presence of entangled motifs against the reference of the general protein case [35], which contains mostly soluble globular proteins. We built a non-redundant data set of membrane protein domains, annotated with the monotopic/transmembrane and peripheral/integral labels. Single domains are the most natural units to investigate self-entanglement of the protein chain, since inter-domain entanglement is unlikely. Moreover, the conservation of the domain length distribution across different data sets (see Figure 2c) allows for an unbiased comparison of the entanglement properties [22].

Our first main result is that entangled loops appear frequently in membrane protein domains as well, albeit less than in the reference set, with one-fifth of transmembrane domains and one-fourth of monotopic domains hosting at least one entangled motif (see Figure 2b). This is unexpected in our view since the folding of membrane proteins already faces the issue of the insertion of the membrane-interacting segments in the correct fashion. The presence of entangled motifs is likely to provide further constraints to the folding process. How this may be accomplished in the bio-genesis of membrane proteins is an intriguing question.

Furthermore, we studied in detail the properties of the entangled motifs detected in membrane protein domains. Surprisingly, the main features of the distribution of the entanglement indicator values are similar to the reference case of general proteins (see Figure 1a). The same distribution is conserved across different organisms (see Figure 1b). This is the first hint that the properties of entangled motifs might be related to the machineries that mediate the bio-genesis of membrane proteins. The latter is one of the most ancient biological processes, and therefore the related machineries are exceptionally broadly conserved [59]. However, there are other properties of the entangled motifs, in addition to their occurrence, that discriminate membrane proteins with respect to the reference set of general proteins. In this respect, we find transmembrane domains consistently more distant from the reference set than monotopic domains (see Figure 2a,b), a trend nicely reflecting the amount of interaction with the membrane.

Our second main result is that the chirality preference for multiple winding entangled motifs changes the membrane proteins with respect to the reference case. The chirality bias might correlate with a bias on the position of the thread with respect to the loop along the chain. In fact, for both membrane proteins and the reference set, we observe for single winding motifs a bias favoring threads with positive chiralities on the N-terminal side of the loop (see Figure 4). This confirms previous results obtained for the reference set [35]. Instead, the favored chirality observed for double-winding entangled motifs is reversed (negative) in the reference set, while remaining positive for transmembrane domains. The statistics of double winding entangled motifs for monotopic domains are too poor to allow any definite conclusion, yet they seem again to behave between transmembrane domains and the reference set (see Figure 4).

The issue of what determines the chirality of global topological arrangements, such as the non-covalent lasso-like entangled motifs discussed here, is not a trivial one. Local chirality biases due, for example, to secondary structure elements, might play a role. Why, then, is the sign of the preferred chirality flipped in the reference set when going from single-winding to double-winding motifs? We speculate that the chirality bias we observe is due to the constraints exerted on the nascent chain by the co-translational folding machinery. For example, it was suggested that the protein backbone is under torsional stress during co-translational folding [60]. Furthermore, we hypothesize that the constraints exerted on single-winding entangled motifs are mainly due to ribosome features, such as the geometry of the exit tunnel; this would explain why the chirality bias is the same for both membrane domains and the reference set. The mouth of the ribosome exit tunnel is positioned ≈35 amino acids from the peptidyl-transferase center inside the ribosome, which might be a sufficient length for entangled loops to form single-winding motifs.

On the other hand, we expect that longer loops are needed to accommodate double-winding motifs; in this case, the constraints exerted on the nascent chain would also be due to the co-translational folding machinery sitting at the ribosome surface. Since the co-translational machinery is not the same for membrane and globular proteins, this might explain the different chirality biases observed. Different bio-genesis pathways are possible for membrane proteins depending on their type and biophysical properties [59], yet for multi-pass transmembrane proteins, a ribosome-bound multi-pass translocon has been characterized, which spans through the ER membrane as well [61]. The membrane itself might then be a factor affecting the chirality preference of double-winding entangled motifs. The need for the nascent transmembrane segments to access the hydrophilic vestibules or the lateral gates of different translocon subunits, prior to translocation across the ER membrane, might impose specific constraints on the chirality of double-winding entangled motifs.

The relationship that we propose here between the chirality bias observed for entangled motifs and the constraints exerted on the nascent chain by the co-translational machinery is at this stage a speculation. It could be supported by coarse-grained simulation of co-translational folding, which should model, even in a simplified way, the presence of the translocon machinery, and by a finer analysis of the properties of entangled motifs in membrane proteins, distinguishing, for example, between type I, type II, and type III membrane proteins.

We also find several examples of membrane protein domains where entangled motifs with opposite chiralities coexist (see Figure 3). One such example is shown in Figure 6, where the entangled motifs span both the membrane and the periplasm. Other examples of entangled membrane protein domains show that entangled motifs can be formed in the cytosol (see Figure 5b) or in the membrane (see Figure 5a), highlighting how they can be formed in different environments. The variety of contexts in which entangled motifs are found makes their relationship with the biological function a subtle one. A reasonable speculation is that the mutual inter-winding of the loop and the thread may, in general, provide some enhanced mechanical stability.

Finally, since entangled motifs are not rare in membrane protein domains, the question of the possible presence of misfolded misentangled species can be raised [45]. Such species could be entangled when the native structure is not, or not entangled when the native structure is, or host-entangled motifs with different properties, for example, opposite chirality, with respect to the native structure [45]. In the context of globular proteins, it was shown that those misentangled species can be populated during or just after translation and that are otherwise similar to the native population and, therefore, able to escape the protein quality control system [44]. On the other hand, those species display a reduced biological activity [46]. It is plausible that a similar scenario might occur for membrane proteins since their bio-genesis employs quality control mechanisms and factors (such as intra-membrane chaperons [59]) in a way non-dissimilar to globular proteins.

## 4. Materials and Methods

### 4.1. Membrane Protein Dataset

We compile a dataset of experimentally resolved chain structures that have been detected to interact with cell phospholipid bilayers, such as the plasma membrane, the endoplasmatic reticulum, or the mitochondrial membrane. Membrane-interacting polypeptide chains are collected from the following databases: MemProtMD [48], mpstruc [5], OPM [49,50], and PDBTM [51,52]. The RCSB Protein DataBank [6,53] is used as an aggregator to retrieve tridimensional structures and membrane-related annotations, such as source organisms, functions, and classifications, coming from the above-mentioned databases. Therefore, among all of the PDB structures stored in the RCSB database, we select those chains (or entities) that have been marked as membrane proteins by at least one of the membrane protein databases. The database was accessed on 1 March 2023, resulting in 28,151 chains.

Polypeptide chains were classified as transmembrane, or equivalently bitopic or monotopic as follows: if the entity was annotated by PDBTM or the “transmembrane” keyword is mentioned in the mpstruc or OPT annotations, then it is classified as transmembrane. On the other hand, if the mpstruc or OPT annotations contain the “monotopic” keyword, then it is classified as monotopic. Chains classified by PDBTM or MemProtMD, or having at least one keyword among “transmembrane”, “intrinsic”, or “integral” mentioned in mpstruc or OPT annotations were considered intrinsic. However, chains that had the “peripheral” keyword in mpstruc or OPT annotations were considered peripheral. A set of 20 chains was manually classified due to conflicting annotations. The classification table is available at https://doi.org/10.25430/researchdata.cab.unipd.it.00000897 (accessed on 19 May 2023). Entries classified as “unclassified” belong to a small set that was undetermined but still considered for the topological analysis.

The CATH database [47] in its 4.3 version was used to subdivide chains into domains. Domains composed of multiple fragments, that is, composed of multiple non-consecutive sequence portions, were excluded, as well as domains with missing regions longer than 10 amino acids. Finally, to reduce redundancy in the dataset, a 35% homology filter is applied, favoring structures with better resolution. After discarding error-prone PDB files, the final dataset, recapitulated also in Table 1, contained 1378 domains. 871 were classified as transmembrane, 494 were monotopic, and 13 were unclassified. Furthermore, 871 were intrinsic, 490 were peripheral, and 17 were unclassified. Appendix A summarizes the pipeline.

The 16,709 protein domain structures in the reference set were obtained after discarding 259 error-prone entries from the 16,968 structures originally reported in a previous analysis [35].

Interaction networks, or contact maps, were computed on the experimentally resolved structures in the same way as in previous works [35,41]. Two residues were in contact if two non-Hydrogen atoms, one in each residue, were found closer than 4.5 Å.

### 4.2. Gaussian Entanglement and Modeling

The linking number *G* between two closed oriented curves $\gamma_{i}=\left\{r_{i}\right\}$ and $\gamma_{j}=\left\{r_{j}\right\}$ in $R^{3}$ can be defined through Gauss’ double integrals [36]:(1)$$\begin{array}{c} G\left(\gamma_{i},\gamma_{j}\right):=\frac{1}{4\pi }\oint_{\gamma_{i}}\oint_{\gamma_{j}}\frac{r_{i}-r_{j}}{{\left|r_{i}-r_{j}\right|}^{3}}\cdot \left(dr_{i}\times dr_{j}\right)\;. \end{array}$$This number is an integer and a topological invariant. A generalization for discrete and open curves is the Gaussian entanglement $G^{\prime }$ [32,33,35]. For a chain with *N* monomers, $\gamma ={\left\{r_{k}\right\}}_{k=1}^{N}$, let $\gamma_{i}={\left\{r_{i}\right\}}_{i=i_{1}}^{i_{2}}$ and $\gamma_{j}={\left\{r_{j}\right\}}_{j=j_{1}}^{j_{2}}$ be two non-overlapping portions of γ. We require $j_{2}-j_{1}\geq m_{j}$ and $i_{2}-i_{1}\geq m_{i}$. In the present work, we choose $m_{i}=4$ and $m_{j}=10$ [35]. In coarse-grained protein chains, $r_{k}$ represents the alpha carbon position vector. Let $R_{i}=\left(r_{i+1}+r_{i}\right)/2$ be the mean position between two subsequent alpha carbons and $\Delta R_{i}=r_{i+1}-r_{i}$ be the virtual bond vector. The Gaussian entanglement, actually describing the self-entanglement between two different portions of the same chain, is defined as:(2)$$\begin{array}{c} G^{\prime }\left(\gamma_{i},\gamma_{j}\right):=\frac{1}{4\pi }\sum_{i=i_{1}}^{i_{2}-1}\sum_{j=j_{1}}^{j_{2}-1}\frac{R_{i}-R_{j}}{{\left|R_{i}-R_{j}\right|}^{3}}\cdot \left(\Delta R_{i}\times \Delta R_{j}\right) \end{array}$$ Crucially, $G^{\prime }\left(\gamma_{i},\gamma_{j}\right)$ is a real number.

In the present contribution, we considered $\gamma_{i}$ as a “loop”, a chain portion “closed” by a non-covalent interaction, with residues $i_{1}$ and $i_{2}$ in contact with each other (see Section 4.1 for details about contact map definition). No similar constraints were imposed on $\gamma_{j}$ and we called it a “thread” if it is involved in an entangled motif.

For any loop $\gamma_{i}$, or equivalently for any native contact, we selected the thread most likely entangling with $\gamma_{i}$ by maximizing $\left|G^{\prime }\left(\gamma_{i},\gamma_{j}\right)\right|$. We can limit the search considering only $\gamma_{j}$ on the *N* (*C*) terminal side of the loop they entangle with, namely N-threads (C-threads). Hence, each loop has a Gaussian entanglement with an N- and a C-thread.

By maximizing $\left|G^{\prime }\left(\gamma_{i}\right)\right|$ over all loops $\gamma_{i}$, one can associate two entanglement indicators to the whole chain configuration. One considering N-threads only and one using C-threads only, $G_{N}^{\prime }$ and $G_{C}^{\prime }$, respectively. The maximum Gaussian entanglement, together with the corresponding loop and thread pair, corresponds to the maximum modulus between these two. This procedure results in the entanglement indicator used in previous works [33,35]. Identification of loops and computation of the Gaussian entanglement were performed using the in-house developed Python package pyge, available at https://github.com/gentangle/pyge (accessed on 19 May 2023), software version number 0.8.0, created by Leonardo Salicari, Padova, Italy.

For the Gaussian entanglement calculation, domains that present a gap shorter than or equal to 10 amino acids in the experimentally resolved structure were completed using the MODELLER software [54,55], software version number 10.4, created by Andrej Sali, San Francisco, CA, USA. Information regarding missing residues was extracted from the PDB header (REMARK 465 and SEQRES) and alignment between the complete sequence and the resolved one was provided to the software for the modeling. One model per missing segment was generated while keeping residues farther than 3 residues from both extremes of the modeled region fixed. Tails were not modeled to avoid spurious backbone entanglements. All scripts used for the present work, and the generated entanglement data, are available at https://doi.org/10.25430/researchdata.cab.unipd.it.00000897 (accessed on 19 May 2023).

## Figures and Tables

**Figure 1 ijms-24-09193-f001:**
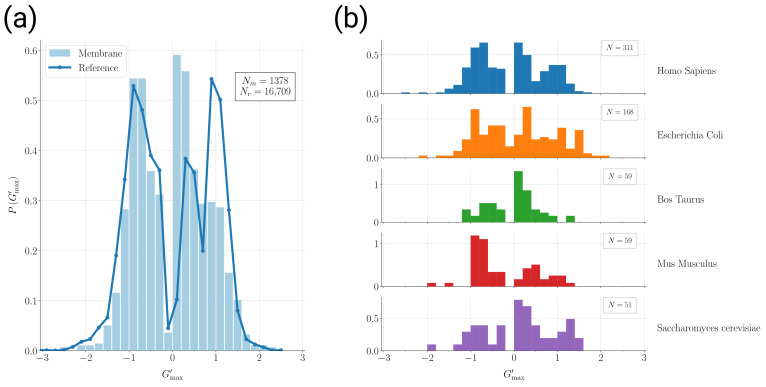
Distribution of the values of the overall entanglement indicator $G_{\max}^{\prime }$ for different sets of protein domain structures: (**a**) $G_{\max}^{\prime }$ probability distributions for 1378 membrane protein domains (light blue filled bars) and for the reference set of 16,709 protein domain structures (dark blue solid line). (**b**) $G_{\max}^{\prime }$ probability distributions for the subsets of the 1378 membrane protein domains corresponding to some of the most represented organisms.

**Figure 2 ijms-24-09193-f002:**
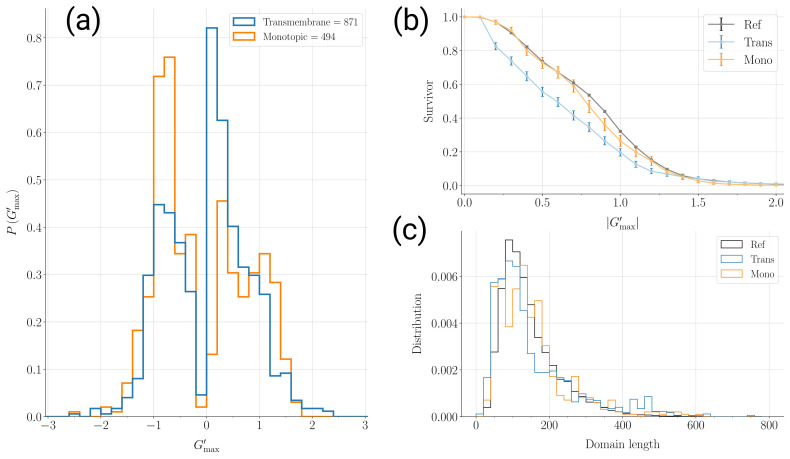
Comparison between monotopic and transmembrane sets of protein domain structures: (**a**) $G_{\max}^{\prime }$ probability distributions for 871 transmembrane protein domains (light blue) and for 494 monotopic protein domains (orange). (**b**) Survival function showing the fraction of structures in the data set with an entanglement indicator greater than $\left|G_{\max}^{\prime }\right|$. Black: Reference set. Light blue: transmembrane protein domains. Orange: Monotopic protein domains. Error bars refer to the 5% and 95% percentiles after 10,000 bootstrap samplings. (**c**) Distributions of the protein domain length for the different data sets. Black: Reference set. Light blue: transmembrane protein domains. Orange: Monotopic protein domains.

**Figure 3 ijms-24-09193-f003:**
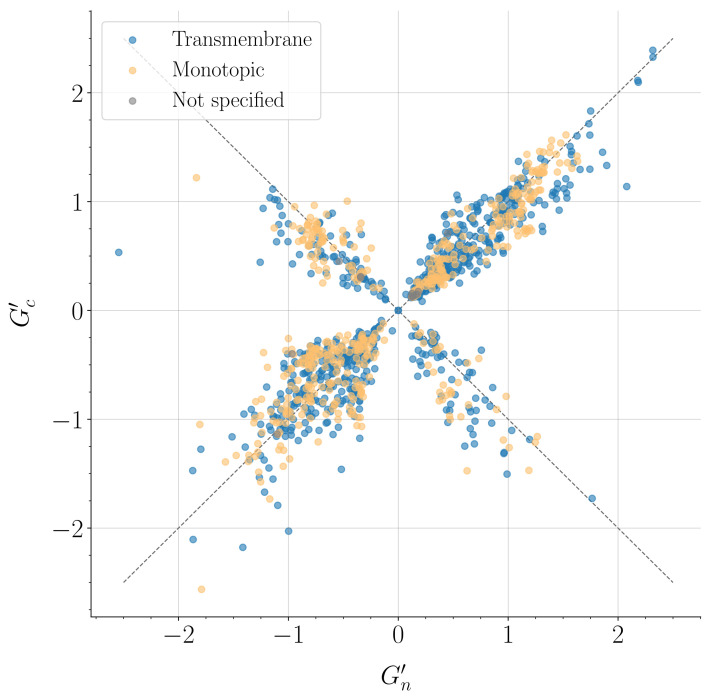
Scatter plot of $G_{N}^{\prime }$ and $G_{C}^{\prime }$ values. Each point in the plot refers to a single domain in the data sets. Light blue: 871 transmembrane protein domains. Orange: 494 monotopic protein domains. Grey: 13 not classified membrane protein domains. Dash lines refer to the the diagonal ($G_{C}^{\prime }=G_{N}^{\prime }$) and the anti-diagonal ($G_{C}^{\prime }=-G_{N}^{\prime }$) lines in the ($G_{N}^{\prime }$,$G_{C}^{\prime }$) plane.

**Figure 4 ijms-24-09193-f004:**
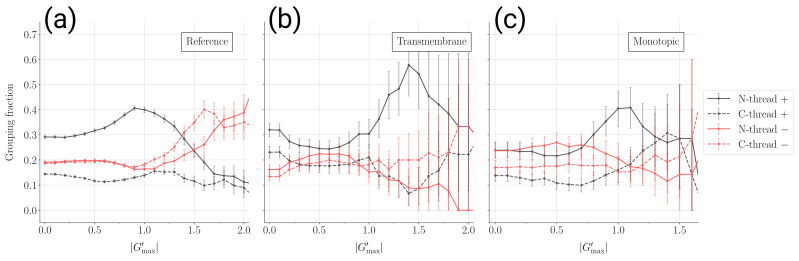
Grouping fractions for different kinds of entangled motifs. The data report the fraction of the survived structures (i.e., with entanglement indicator greater than $\left|G_{\max}^{\prime }\right|$, see Figure 2b) that belong to a given group. In all groups, we consider only structures for which the most entangled N-thread and the most entangled C-thread share the same chirality. Solid black: N-threads with positive chirality ($G_{N}^{\prime }>G_{C}^{\prime }>0$). Dashed black: C-threads with positive chirality ($G_{C}^{\prime }>G_{N}^{\prime }>0$). Solid red: N-threads with negative chirality ($G_{N}^{\prime }<G_{C}^{\prime }<0$). Dashed red: C-threads with negative chirality ($G_{C}^{\prime }<G_{N}^{\prime }<0$). Error bars refer to the 5% and 95% percentiles after 10,000 bootstrap samplings: (**a**) Reference set. (**b**) transmembrane protein domains. (**c**) Monotopic protein domains.

**Figure 5 ijms-24-09193-f005:**
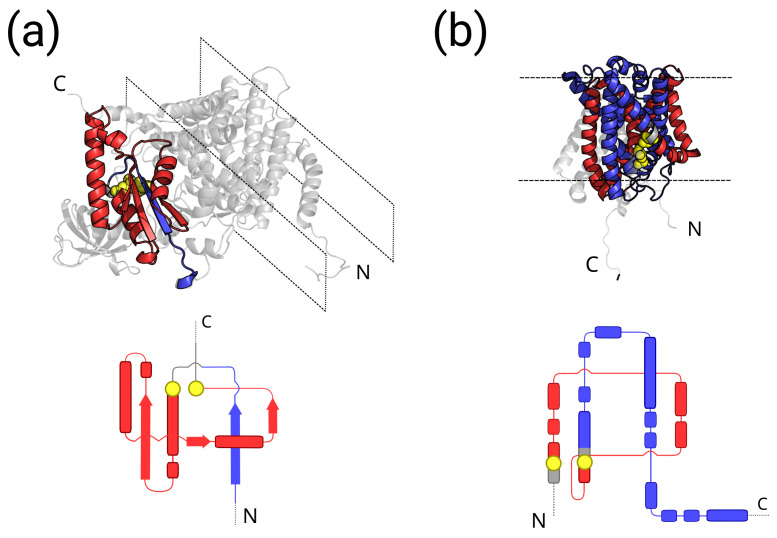
Examples of transmembrane proteins with entangled motifs. **Top**: Cartoon structures. **Bottom**: Temptative planar structure projection to better appreciate the winding of the thread around the entangled loop and the corresponding chirality; secondary structure elements are shown as a guide. Blue: thread. Red: entangled loop. Grey: other portions of the protein chain. Yellow: residues at the loop ends that close the non-covalent lasso; all heavy atoms are shown in the cartoon structure for the yellow residues. (**a**) A subunit from the bacterial cellulose synthase (Bcs) 4P02 complex with a positive chirality N-thread entangled motif ($G_{N}^{\prime }=1.10>G_{C}^{\prime }=0.79$); membrane boundaries are shown as dashed black lines. (**b**) APC transporter 3GIA with a positive chirality C-thread double winding entangled motif ($G_{N}^{\prime }=2.32<G_{C}^{\prime }=2.39$); membrane boundaries are shown as dashed black lines.

**Figure 6 ijms-24-09193-f006:**
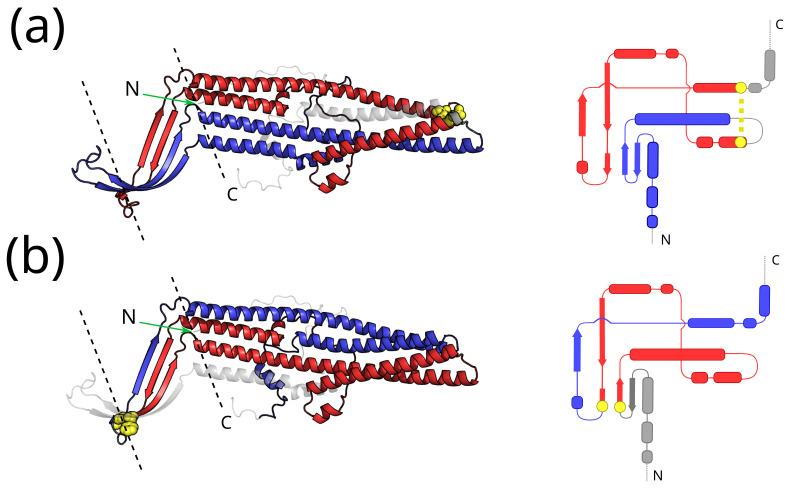
Example of a transmembrane protein with coexisting entangled motifs with opposite chiralities. **Left**: Cartoon structure of the A chain from the CmeC outer membrane channel homo-trimer 4MT4 from *Campylobacter jejuni*; membrane boundaries are shown as dashed black lines. **Right**: Temptative planar structure projection to better appreciate the winding of the thread around the entangled loop and the corresponding chirality; secondary structure elements are shown as a guide. Blue: thread. Red: entangled loop. Grey: other portions of the protein chain. Yellow: residues at the loop ends that close the non-covalent lasso; all heavy atoms are shown in the cartoon structure for the yellow residues, whereas the non-covalent contact is shown as a yellow dotted line in the planar projection. (**a**) Positive chirality N-thread ($G_{N}^{\prime }=1.20$). (**b**) Negative chirality C-thread ($G_{C}^{\prime }=-1.19$).

**Table 1 ijms-24-09193-t001:** Statistics of membrane protein domains. Domains are subdivided with respect to how they interact with the membrane, with only one side of the phospholipid bilayer (monotopic) or both (transmembrane), and the characteristic time of interaction, long (integral) or transient (peripheral). Each domain is labeled using the classification of the chain it belongs to. Unclassified domains correspond to those that can not be unambiguously assigned to one of the other classes.

**Domains**	**Transmembrane**	**Monotopic**	**Unclassified**
1378	871	494	13
**Domains**	**Integral**	**Peripheral**	**Unclassified**
1378	871	490	17

## Data Availability

All scripts used for the present work are available at https://doi.org/10.25430/researchdata.cab.unipd.it.00000897 (accessed on 19 May 2023).

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
