# Peer review of "Entangled Motifs in Membrane Protein Structures"

_ijms, 2023, doi:10.3390/ijms24119193_

Round 1

Reviewer 1 Report

This is an excellent study by the Travato lab, who discovered the wide spread presence of non-covalent lasso entanglements in proteins. In this research they focus on characterizing the distribution and properties of non-covalent lasso entanglements in membrane proteins. This is study is novel, as the hypothesis is that the membrane environment the protein exists in, as well as the co-translational insertion process of the protein in the membrane, may bias the distribution of this novel class of tertiary structural motifs. Through a rigorous structural analysis of membrane protein structures, they indeed find that 20% of membrane protein domains contain these motifs (compared to 33% of globular proteins), and that there are interesting patterns of chirality in these entangled proteins relative to the globular protein set. The authors reasonably argue that the patterns of these chirality switches are likely related to constraints arising during co-translational protein folding.

Overall, this study advances the field of protein structural biology with regard to membrane proteins, and is likely to be of general interest.

There are some minor wording issues I caught, although the authors should carefully check for others before accepting for publication, for example:

The sane distribution”  “The same distribution”

Minor edits to improve English in just a few spots.

Author Response

We are grateful to the reviewer for her/his positive assessment of our manuscript.

There are some minor wording issues I caught, although the authors should carefully check for others before accepting for publication, for example:

“The sane distribution” -> “The same distribution”

We apologize for the several wording issues / typos that were indeed present in the manuscript. As suggested by the reviewer, we checked carefully for them and we hope we succeeded in revising them.

Reviewer 2 Report

Salicari and colleagues reported a computational analysis of entangled motifs in membrane protein. This work shows the comparison of entangled motif distribution in both membrane proteins and single domain proteins and analyzed the bias of chiralities of N- and C-threads. These findings are important for deepening the understandings of membrane protein folding mechanism. The manu is well-written in general and may need minor revisions to improve the quality before publication.

Comments:

What is the relationship between the entangled motif and the biological function? Please provide some general insights based on the example proteins listed in this work.

Figure 1, it looks to me that for membrane protein, the positive and negative tails are more or less symmetric, so I would suggest change the statement in line 181 and 182. And in B, the authors listed 5 species but four of them are mammalian. You may consider adding other species such as yeast, C. elegans, Drosophila to make a comparison among wider species.

Figure 2, the light green bar to me is confusing and blocks the details of the other two distributions. I would suggest to plot it as a line or other better ways to show it.

 Please double check the Figure 5 legend. The description for b should be a.

Author Response

We thank the reviewer for her/his positive assessment of our manuscript and for her/his suggestions that helped us to considerably improve the manuscript.

Comments:

What is the relationship between the entangled motif and the biological function? Please provide some general insights based on the example proteins listed in this work.

The question raised by the reviewer is a very interesting one. We feel at this stage it is hard to provide a general insight on the relationship with the biological function. One can only guess that the mutual inter-winding of the loop and the thread can in general  provide some enhanced mechanical stability. Moreover, the specific examples that we show exhibit entangled motifs either fully in the cytosol (Fig. 5a in the revised manuscript), or fully within the membrane (Fig. 5b in the revised manuscript), or spanning both the membrane and the periplasm. The variety of contexts in which entangled motifs are found makes the relationship with the biological function a subtle one. We added two sentences addressing this issue according to the above lines in the Discussion section of the revised manuscript.

Figure 1, it looks to me that for membrane protein, the positive and negative tails are more or less symmetric, so I would suggest change the statement in line 181 and 182. And in B, the authors listed 5 species but four of them are mammalian. You may consider adding other species such as yeast, C. elegans, Drosophila to make a comparison among wider species.

We thank the reviewer for her/his suggestions. We revised the sentence about distribution tails of Fig. 1A accordingly; we now state that a bias favoring negative over positive chirality for deeply entangled structures is present only in the reference set. We also replaced the Rattus Norvegicus entanglement distribution in Fig. 1B with the Saccharomyces Cerevisiae one. The conclusions that we can draw remain the same, so that we only revised the related caption. The limiting factor in selecting organisms for this analysis is the available statistics.

Figure 2, the light green bar to me is confusing and blocks the details of the other two distributions. I would suggest to plot it as a line or other better ways to show it.

We thank the reviewer for her/his observation. We changed the plot using lines as suggested by the reviewer.

Please double check the Figure 5 legend. The description for b should be a.

The reviewer is definitely right, the caption was inconsistent with the figure. In fact, in the revised manuscript we exchanged the two panels in Figure 5 so that the caption is now correct. We changed the labels in the main text, accordingly. This way, Figure 5a is the one discussed earlier in the manuscript.